# Multiple Myeloma: The Role of Autologous Stem Cell Transplantation in the Era of Immunotherapy

**DOI:** 10.3390/cells13100853

**Published:** 2024-05-16

**Authors:** Serena Rocchi, Beatrice Anna Zannetti, Giovanni Marconi, Francesco Lanza

**Affiliations:** Hematology Unit and Romagna Metropolitan Transplant Network—Ravenna, University of Bologna, 40126 Bologna, Italy; serena.rocchi@auslromagna.it (S.R.); beatriceanna.zannetti@auslromagna.it (B.A.Z.); giovanni.marconi@unibo.it (G.M.)

**Keywords:** multiple myeloma, stem cell transplantation, immunotherapy, bispecific antibodies

## Abstract

Upfront high-dose therapy with melphalan (HDM) followed by autologous stem cell transplantation (ASCT) has established itself as a core treatment for newly diagnosed multiple myeloma (NDMM) patients in the past 30 years. Induction therapy, HDM-ASCT, and subsequent consolidation and maintenance therapy comprise the current fundamental framework for MM treatment. The introduction of anti-CD38 monoclonal antibodies such as daratumumab and isatuximab has changed the treatment paradigm for transplant-eligible NDMM patients in that quadruplets have become the new standard induction therapy. The treatment landscape of MM is undergoing a transformative shift with the introduction of potent new immunotherapies, such as chimeric antigen receptor (CAR)-T cells and bispecific antibodies (BsAbs), which are currently used in the relapsed/refractory setting (RRMM) and are already being tested in the NDMM. This review will focus on the incorporation of immunotherapy in the treatment scenario of NDMM patients eligible for ASCT.

## 1. Introduction

Multiple myeloma (MM) is the third most common hematological malignancy; it is a clonal plasma cell dyscrasia, accounting for 10% of newly diagnosed hematologic malignancies. Over the past two decades, the expansion of treatment options, including the introduction of targeted therapies and immunomodulatory drugs, has dramatically improved patient outcomes in terms of response rates and survival.

Despite these advancements, autologous stem cell transplantation (ASCT) following high-dose melphalan (HDM) continues to be the standard of care for eligible patients with newly diagnosed MM, showing high response rates and extended survival times. The therapeutic path includes four phases: induction, HDM plus ASCT, consolidation, and maintenance, a model associated with high response rates, prolonged progression-free (PFS), and overall (OS) survival [1,2,3].

The latest 2021 European Hematology Association and European Society for Medical Oncology (ESMO) guidelines recommend induction therapy with VTd (bortezomib, thalidomide, dexamethasone) or VRd (bortezomib, lenalidomide, dexamethasone) plus the anti-CD38 monoclonal antibody daratumumab, followed by HDM-ASCT and lenalidomide maintenance [4,5].

The role of ASCT may continue to evolve with the introduction of immunotherapies in a front-line setting.

Immunotherapy in MM has been widely explored in recent years [6,7].

Currently, there are two different approaches, aiming at either passive or active reactivation of the immune system [8]. Passive immunotherapy includes monoclonal antibodies (MoAbs), both “naked” and conjugated to a cytotoxic agent. Modern strategies of active immunotherapy are represented by bispecific antibodies (BsAbs) and cellular therapy with chimeric antigen receptor (CAR)-T cells, which activate and redirect the T-cell compartment of the immune system against the neoplastic cells, and which have recently revolutionized the therapeutic landscape of MM [9,10]. In both cases, these therapeutic options target surface antigens predominantly expressed on MM cells, with a role in the proliferation and survival of neoplastic plasma cells (PCs), and less represented on other cell types, thus minimizing off-target effects [9].

These immune-based approaches have shown notable anti-myeloma effects with deep and durable responses in initial clinical trials of heavily pretreated MM patients. Their use in the earlier stages of the disease has produced impressive results, raising the question of how to integrate these drugs into the therapeutic algorithm of transplant-eligible NDMM.

## 2. Current Consideration on the Role of ASCT in MM

In transplant-eligible NDMM patients, HDM plus ASCT remains the standard of care recommended by international guidelines such as those of the American Society of Clinical Oncology (ASCO), ESMO, and European Bone Marrow Transplantation (EBMT) [4,11,12,13]. Until a few years ago, regimens for induction therapy commonly included a proteasome inhibitor (PI), an immunomodulatory drug (IMiD), and dexamethasone (a three-drug therapy—triplet). More recently, the introduction of anti-CD38 monoclonal antibodies, daratumumab and isatuximab, has changed the treatment paradigm for transplant-eligible NDMM patients in that quadruplets have supplanted triplets [14,15].

### 2.1. Daratumumab-Based Induction Regimens

Daratumumab is a human IgG/kappa MoAb directed against targeting CD38 antigen surfaces with different mechanisms of action [16,17]. Daratumumab has been approved for use in combination with various regimens for the treatment of RRMM [18,19] and newly diagnosed transplant-ineligible MM [20,21,22].

The addition of daratumumab to standard triplets as induction and consolidation in transplant-eligible NDMM has been extensively evaluated in many randomized controlled clinical trials (Table 1), and now it represents a new standard of care.

The phase 3 CASSIOPEIA trial marked the first evidence of the clinical advantage of incorporating daratumumab into the standard VTd regimen [23]. This study involved an initial randomization of participants to receive either VTd alone or in combination with daratumumab (D-VTd) across four cycles of pre-transplant induction and two cycles of post-transplant consolidation. Subsequently, responders were randomized again to either continue with maintenance daratumumab or switch to observation. The D-VTd regimen, administered both before and after ASCT, enhanced the primary outcomes of stringent complete response (sCR) and PFS compared to VTd alone. Additionally, 64% of D-VTd patients achieved minimal residual disease (MRD) negativity by day 100 post-transplant, compared to 44% of those on VTd (*p* < 0.0001). These findings led to the FDA and EMA’s approval of the four-drug combination D-VTd. Notably, CASSIOPEIA was also the first trial to demonstrate the benefits of daratumumab maintenance over observation following ASCT [24]. Patients who continued daratumumab treatment beyond two years post-consolidation experienced a median PFS that surpassed that of the observation group (not reached vs. 46.7 months; HR = 0.53, *p* < 0.0001). While the PFS advantage was apparent in the VTd group with daratumumab maintenance compared to VTd with observation only (HR = 0.32, *p* < 0.0001), no significant PFS difference was found between the D-VTd induction/consolidation with daratumumab and D-VTd with observation (HR = 1.02, *p* = 0.91). Moreover, the daratumumab group showed higher rates of complete response (CR) or better (73% vs. 61%, *p* < 0.0001), improved responses (62% vs. 47%, *p* < 0.0001), MRD negativity (assessed by next-generation sequencing at 10^−5^; 59% vs. 47%, *p* = 0.0001), and conversion to MRD negativity (44% vs. 30%, *p* = 0.0004) compared to the observation group.

The phase II GRIFFIN study rigorously assessed the impact of adding daratumumab to the VRd regimen (bortezomib, lenalidomide, and dexamethasone) in NDMM [25]. This randomized trial involved 207 NDMM patients who were divided into groups to receive either D-VRd or VRd for 4 cycles of induction, followed by ASCT, 2 cycles of consolidation with D-VRd or VRd, and 26 cycles of maintenance therapy with either lenalidomide alone or combined with daratumumab. After a median follow-up of 13.5 months, the primary outcome of stringent complete response sCR post-consolidation was reached by 42.4% of patients in the D-VRd group compared to 32.0% in the VRd group (*p* = 0.068). Furthermore, the D-VRd group showed superior secondary outcomes, with an ORR of 99.0% compared to 91.8% in the VRd group (*p* = 0.0160) and a higher rate of VGPR or better (90.9% vs. 73.2%, *p* = 0.0014). Over time, these responses deepened, with 62.6% of the D-VRd group achieving sCR by the last follow-up at 22.1 months, compared to 45.4% in the VRd group (*p* = 0.0177). Additionally, the rate of complete response (CR) or better was 79.8% in the D-VRd group versus 60.8% in the VRd group (*p* = 0.0045). At this follow-up point, the MRD-negativity rate was significantly higher in the D-VRd group than in the VRd group (51.0% vs. 20.4%, *p* < 0.0001), especially among patients achieving CR or better (62.0% vs. 32.2%, *p* = 0.0006). Although the median PFS was not reached, the final analysis at 49.6 months demonstrated a 55% reduction in the risk of disease progression (PD) or death in the D-VRd arm compared to the VRd arm (HR, 0.45; 95% CI, 0.21–0.95; *p* = 0.0324). Importantly, sCR rates increased to 67% for D-VRd and 48% for VRd after two years of maintenance (*p* = 0.0079). Given the positive trends in PFS and the high rates of sCR and MRD negativity, the D-VRd quadruplet has become a standard treatment in the US. However, the limitations of the GRIFFIN study, a phase II trial with nearly 200 participants not powered to definitively assess PFS, suggest that further confirmation through larger phase 3 trials is necessary [26].

The efficacy of the D-VRd regimen has been recently confirmed by the phase 3 PERSEUS study [27]. 709 transplant-eligible patients with NDMM were randomized to receive either subcutaneous daratumumab plus VRd induction and consolidation regimen and with lenalidomide maintenance (D-VRd group) or VRd induction and consolidation regimen and maintenance with lenalidomide alone (VRd group). Interestingly, in the PERSEUS trial, patients who achieved a CR or better and maintained MRD-negativity (sensitivity at ≤10^−5^) for at least 12 months stopped daratumumab after 24 months of maintenance and continued with only lenalidomide until disease relapse. At a median follow-up of 47.5 months, PD or death had occurred in 50 of 355 patients (14.1%) in the D-VRd group and 103 of 354 patients (29.1%) in the VRd group. The estimated 48-month PFS rate was 84.3% for the D-VRd group compared to 67.7% for the VRd group (HR 0.42, 95% CI, 0.30 to 0.59; *p* < 0.001). The D-VRd group also showed higher rates of CR or better (87.9% vs. 70.1%, *p* < 0.001) and MRD-negative status (75.2% vs. 47.5%, *p* < 0.001), with 64.8% maintaining MRD-negative status for at least 12 months compared to 29.7% in the VRd group. By the time of analysis, 207 of the 322 patients in the maintenance phase of the D-VRd group had stopped daratumumab as per the protocol. These findings affirm the benefits of incorporating daratumumab into the standard treatment regimen for NDMM.

MRD negativity after treatment is linked to improved outcomes in NDMM patients, though its role in guiding therapy adjustments is still being established. In the phase 2 MASTER trial, NDMM patients eligible for transplant underwent four cycles of daratumumab, carfilzomib, lenalidomide, and dexamethasone (Dara-KRd) induction therapy, followed by HDM-ASCT, and up to two cycles of consolidation with Dara-KRd [28]. The primary goal was to achieve MRD negativity (<10^−5^). Participants who achieved MRD negativity after or during two subsequent cycles switched to observation with ongoing MRD monitoring, while those without two consecutive MRD-negative determinations continued with lenalidomide maintenance. Of the 123 evaluable patients, 53 (43%) had no high-risk cytogenetic aberrations (HRCAs), 46 (37%) had one HRCA, and 24 (20%) had multiple HRCAs. MRD negativity was achieved in 81% of the study cohort. Eighty-four patients met the criteria for MRD-SURE and entered protocol-directed observation, whereas 24 received ongoing standard lenalidomide maintenance. With a median follow-up of 42.2 months, the 36-month PFS was 88%, 79%, and 50% for participants with no, one, and two or more HRCAs, respectively. This trial exemplifies the potential of MRD-driven therapy to offer a treatment-free state, highlighting the need for further improvements in the prognosis for patients with ultra-high-risk MM.

### 2.2. Isatuximab-Based Induction Regimens

Isatuximab is a chimeric IgG monoclonal antibody that specifically targets a unique epitope on CD38, exerting its anti-myeloma effects through multiple mechanisms. These include antibody-dependent cellular cytotoxicity, complement-dependent cytotoxicity, antibody-dependent cellular phagocytosis, direct induction of apoptosis, and inhibition of the CD38 enzyme’s intrinsic activity [29]. Isatuximab has received approval for use in combination with various treatment regimens for RRMM and is also being explored in the initial treatment of transplant-eligible NDMM patients (Table 1) [30,31,32,33].

**Table 1 cells-13-00853-t001:** Main results of anti-CD38 monoclonal antibodies into ASCT program in newly diagnosed MM patients.

Trial	Phase	Design	Follow Up	sCR (MRD) ^a^	PFS
**CASSIOPEIA** [23,24]	3	VTd vs. D-VTd induction (4 cycles) and consolidation (2 cycles), with D maintenance or observation	Day 100 after ASCT	29% vs. 20% (64% vs. 44%)	NR vs. 47 mo ^b^
**GRIFFIN** [25,26]	2	D-VRd induction (4 cycles) and consolidation (2 cycles) plus DR maintenance or VRd induction (4 cycles) and consolidation (2 cycles) plus R maintenance	50 mo	67% vs. 48% (64% vs. 30%)	87% vs. 70%
**PERSEUS** [27]	3	D-VRd induction (4 cycles) and consolidation (2 cycles) plus DR maintenance ^c^ or VRd induction (4 cycles) and consolidation (2 cycles) plus R maintenance	48 mo	88% vs. 70% ^d^ (75% vs. 48%)	84% vs. 68%
**MASTER** [28]	2	D-KRd induction (4 cycles) and consolidation (2 cycles), with R maintenance or observation ^e^	42 mo	78% vs. 86% vs. 79% ^f^	88% vs. 79% vs. 50% ^f^
**GMMG-HD7** [34]	3	Isa-VRd vs. VRd induction (3 cycles) with maintenance with Isa-R or R	After induction therapy	(50% vs. 36%)	Ongoing ^g^
**GMMG-CONCEPT** [35]	2	Isa-KRd induction (6 cycles) and consolidation (4 cycles) with Isa-KR maintenance ^h^	44 mo	73% (68%) ^i^	NR

sCR: stringent complete remission; MRD: minimal residual disease; PFS: progression free survival; VTd: bortezomib, thalidomide, and dexamethasone; D-VTd: daratumumab, bortezomib, thalidomide, and dexamethasone; D: daratumumab; ASCT: autologous stem cell transplantation; vs: versus; NR: not reached; mo: months; D-VRd: daratumumab, bortezomib, lenalidomide, and dexamethasone; DR: daratumumab and lenalidomide; VRd: bortezomib, lenalidomide, and dexamethasone; R: lenalidomide; D-KRd: daratumumab, carfilzomib, lenalidomide, and dexamethasone; Isa-VRd: isatuximab, bortezomib, lenalidomide, and dexamethasone; Isa-R: isatuximab and lenalidomide; Isa-KRd: isatuximab, carfilzomib, lenalidomide, and dexamethasone; Isa-KR: isatuximab, carfilzomib, and lenalidomide. ^a^: 10^−5^ sensitivity threshold; ^b^: at the follow-up of 35 months; ^c^: after 24 months of maintenance therapy, daratumumab was discontinued in patients who reached CR or better with sustained MRD-negative status (at a sensitivity threshold of ≤10^−5^) for at least 12 months; these patients continued to receive lenalidomide until PD; ^d^: percentage of patients with a complete response or better; ^e^: the treatment was stopped for patients in MRD negativity after or during two consecutive phases; for these patients an observation period with MRD surveillance was started; participants for whom two consecutive MRD-negative determinations were not confirmed received maintenance with lenalidomide; ^f^: Of 123 participants in the MASTER trial for 118 (96%) MRD was evaluable by next-generation sequencing and 84 of them (71%) reached MRD-SURE and treatment cessation; MRD negativity and 36-month PFS were evaluated among all participants and they correspond to patients with no, one and two or more high risk cytogenetic abnormalities, respectively; ^g^: the study is ongoing and will follow-up patients who had transplantation and maintenance, eventually reporting progression-free survival and overall survival results in part 2; ^h^: high-risk MM patients were defined by International Staging System stage II/III combined with del17p, t(4;14), t(14;16), or more than three 1q21 copies as high-risk cytogenetic aberrations; ^i^: at the end of consolidation.

In the phase 3 GMMG-HD7 clinical trial, 660 transplant-eligible NDMM patients were randomized to receive three cycles of induction therapy with either isatuximab plus bortezomib, lenalidomide, and dexamethasone (Isa-VRd) or the standard VRd alone. The primary outcome, assessed by flow cytometry, was MRD negativity in the intention-to-treat population. The trial found that 50% of patients in the isatuximab group achieved MRD negativity after induction, compared to 36% in the control group (*p* = 0.00017) [34].

Furthermore, isatuximab’s efficacy was under evaluation in the phase 2 GMMG-CONCEPT trial, combined with carfilzomib, lenalidomide, and dexamethasone (KRd) in cytogenetically high-risk, both young (≤70 years) and elderly (>70 years) NDMM patients [35]. This study classified patients as high-risk based on the International Staging System (ISS) stages II/III and specific cytogenetic abnormalities such as del17p, t(4;14), t(14;16), or more than three copies of 1q21. These transplant-eligible NDMM patients received Isa-KRd for induction/consolidation and continued with Isa-KR for maintenance. The primary endpoint of the study was achieving MRD negativity at the end of the consolidation phase, and the secondary endpoint was PFS. By the end of consolidation, 72.8% of patients achieved a complete or stringent complete response, and 18.2% reached a very good partial response, with an overall response rate of 94.9%. At this stage, 67.7% of patients were MRD-negative, and 81.8% reached MRD negativity at some point during the treatment. Sustained MRD-negative status for at least 6 and 12 months was achieved by 72 and 62 patients, respectively, corresponding to sustained MRD negativity rates of 72.7% and 62.6%. With a median follow-up of 44 months, the median PFS had not been reached, underscoring Isa-KRd’s potential to significantly impact MRD negativity and improve outcomes in high-risk NDMM populations.

### 2.3. Stem Cell Mobilization and Harvesting

Mobilization of CD34+ stem cells from the bone marrow to the peripheral blood is a prerequisite for harvesting an adequate number of hematopoietic stem cells (HSC). A minimum of 2 × 10^6^/kg CD34+ cells is required to perform ASCT, while the ideal target for 1 ASCT is >3 × 10^6^/kg CD34+ cells, and that for 2 ASCT is >6 × 10^6^/kg CD34+ cells. The optimal stem cell mobilization strategy remains a matter of debate [36,37,38]. Currently, stem cell mobilization can be performed with granulocyte colony-stimulating factor (G-CSF), preceded or not by chemotherapy with Cyclophosphamide (Cy) at a variable dose of 1.5–4 g/m^2^. Recently, plerixafor (scissor enzyme selective and reversible antagonist of CXCR4) has been approved, which is particularly effective in reducing the failure rate of the procedure by preventing the adherence of HSCs to the marrow matrix and thus increasing their release into the circulation. After its approval, the chemotherapy-free protocol has been adopted by many centers because of the less adverse event rates, such as neutropenia or infections.

Recent data seem to indicate that the introduction of anti-CD38 monoclonal antibodies in the induction therapy of NDMM is associated with a lower capacity to mobilize and collect HSC. The biological reasons for these findings have yet to be clarified. An important role could be played by the expression of CD38 on normal bone marrow and mobilized hematopoietic progenitor [39,40,41].

In the phase 3 CASSIOPEIA trial, stem cell collection was notably less efficient following the D-VTd regimen compared to VTd (6.7 vs. 10.0 × 10^−6^/kg), with higher usage of plerixafor (21.7% vs. 7.9%) and increased rates of collection failures (defined as collections <5 × 10^−6^/kg, 24.6% vs. 11.4%) [42]. Further analysis in the phase 2 MASTER and GRIFFIN trials sought to assess the impact of the daratumumab, PI, and IMID combination on hematopoietic stem cell mobilization [43]. Within these studies, up to 97% of D-KRd patients and 72% of the D-VRd cohort needed the use of plerixafor. The median total CD34+ cell yield was 6.0 × 10^−6^/kg in the MASTER trial, 8.3 × 10^−6^/kg for D-VRd, and 9.4 × 10^−6^/kg for VRd in the GRIFFIN study. Among those mobilized, the rate of needing to remobilize was 7% for D-KRd, 2% for D-VRd, and 6% for VRd. Despite the lower yields and increased plerixafor use in the daratumumab-containing regimens in the NDMM setting across the MASTER, GRIFFIN, and CASSIOPEIA trials, the integration of daratumumab into the PI-IMIds triplet therapy did not negatively affect the feasibility and safety of conducting ASCT or hinder successful engraftment. Several retrospective clinical trials have described HSC mobilization and collection in patients receiving daratumumab-based induction regimens [44,45,46].

A German retrospective single-center study evaluated 179 transplant-eligible NDMM patients treated according to the GMMG-HD6 and GMMG-HD7 clinical trials with respect to PBSC mobilization and collection [47]. Patients were grouped according to induction therapy: VRd (6 cycles: 44 patients, 4 cycles: 51 patients), Isa-VRd (35 patients), or elotuzumab-VRd (49 patients). All 179 patients received a chemotherapy-based mobilization. Leukapheresis collection was considered successful if a target dose of >6 × 10^6^/kg CD34+ cells was reached. The addition of isatuximab to VRd had no significant negative impact on HSC mobilization, considering mean peripheral blood CD34^+^ cell count, number of leukapheresis sessions, plerixafor use, and overall CD34+ cell collection results. Multivariable logistic regression analysis confirmed these results.

## 3. T Cell Redirecting Therapy in MM

### 3.1. T Cell Redirecting Therapy Targeting BCMA

As previously mentioned, active immunotherapy works against specific targets on MM cells, activating and redirecting the T cell compartment of the immune system [9,10].

In the search for ideal targets over the years, B-cell Maturation Antigen (BCMA) has emerged as a milestone in MM treatment [48]. BCMA, also known as TNFRSF17, is a member of the tumor necrosis factor receptor (TNFR) superfamily, which binds B-cell activating factor (BAFF) and a proliferation-inducing ligand (APRIL), activating the NF-κB signaling pathway and, consequently, playing a key role in B-cell maturation and differentiation [49,50]. BCMA is mainly expressed on mature B cell surfaces, leading to the survival of long-lived PCs, with minimal expression on other hematopoietic and non-hematopoietic cells and overexpression in MM cells [50,51]. Moreover, BCMA expression increases as the disease progresses, making it a promising therapeutic target [48].

Considering these findings, two new anti-BCMA immunotherapeutic approaches have emerged: CAR-T cells and BsAbs. The results of the main clinical trials are shown in Table 2.

Among anti-BCMA CAR-T cells, idecabtagene vicleucel (ide-cel or bb2121) and ciltacabtagene autoleucel (cilta-cel or JNJ-4528), both with the 4-1BB co-stimulation domain, showed encouraging results in phase 2 and 1b/2 studies for RRMM patients, thus leading to their approval in clinical practice [52,53,54]. In the KarMMa and CARTITUDE-1 trials, the ORR was approximately 73% and 98%, with a high rate of CR and MRD-negativity (CR: 33% and 83%, respectively; MRD-negativity in evaluable patients: 28% and 92%, respectively). Moreover, median PFS and OS were 8.8 months and 19.4 months in the KarMMa trial, respectively [50], and they were not reached in the updated analysis of the CARTITUDE-1 study (27-month PFS: 55%; 27-month OS: 70%) [54]. The most frequent adverse events with ide-cel and cilta-cel were cytokine release syndrome (CRS) (84% and 95%, respectively, mostly of grade 1–2) and hematological toxicity (91% and 96%, respectively, mostly of grade ≥ 3). The incidence of neurological toxicity associated with effector cells of the immune system (ICANS) was low (18% and 17%, respectively, only ≤3% of grade ≥ 3). The main differences in the safety profile between the two constructs were the onset time of CRS (1 day for ide-cel versus 7 days for cilta-cel) and a different kind of neurotoxicity with cilta-cel. This neurological adverse event was mainly characterized by polyneuropathy and movement disorders with a delayed median onset and resolution time (27 days and 75 days, respectively) compared to ICANS, which was observed in a small percentage of patients in the CARTITUDE-1 study (12%, of which 9% were grade ≥ 3) [54]. The pathogenic mechanism of this type of neurotoxicity is not yet fully understood, but it has been hypothesized to be related to the expression of BCMA in the basal ganglia [55,56].

These great results for RRMM patients (with at least 3 previous lines of therapy, including an IMiD, a PI, and an anti-CD38 monoclonal antibody) have laid the basis for further studies in earlier lines of therapy and disease setting, such as KarMMa-2 (NCT03601078), KarMMa-3 (NCT03651128), KarMMa-4 (NCT04196491), CARTITUDE-2 (NCT04133636), CARTITUDE-4 (NCT04181827), CARTITUDE-5 (NCT04923893), and CARTITUDE-6 (NCT05257083) clinical trials.

The phase 3 KarMMa-3 trial focused on patients with RRMM who had previously undergone two to four treatment regimens, including IMiDs, PIs, and daratumumab, and were refractory to their last regimen. Participants were randomly assigned at a 2:1 ratio to receive either ide-cel or one of five standard treatment regimens [57]. The primary outcome measured was PFS. At a median follow-up of 18.6 months, the ide-cel group demonstrated a significantly longer PFS compared to those on standard regimens, with medians of 13.3 months versus 4.4 months, respectively (*p* < 0.001). Treatment with ide-cel also led to a higher response rate, with an overall response rate (ORR) of 71% compared to 42% in the standard regimen group; the percentage achieving stringent complete response (sCR) or complete response (CR) was 39% versus 5%. Additionally, minimal residual disease (MRD) negativity was confirmed in 51 patients (20%) in the ide-cel group compared to just 1 patient (1%) in the standard regimen group. Cytokine release syndrome (CRS) occurred in 88% of the ide-cel-treated patients, primarily in grades 1 or 2 (83%), with grade 3 or higher events in 5% of patients, including two fatalities. Immune effector cell-associated neurotoxicity syndrome (ICANS) was observed in 15% of patients, mostly in grades 1 or 2. The phase 3 CARTITUDE-4 trial targeted patients with multiple myeloma (MM) who were refractory to lenalidomide and had undergone one to three prior treatment lines. Participants were randomized to receive either cilta-cel or the physician’s choice of standard care deemed effective [58]. The primary endpoint assessed was progression-free survival (PFS). The results showed that cilta-cel significantly reduced the risk of PD or death compared to standard care (hazard ratio, 0.26; 95% confidence interval [CI], 0.18 to 0.38; *p* < 0.001). A higher proportion of patients in the cilta-cel group achieved a stringent complete response (sCR) or complete response (CR) compared to those in the standard care group (73.1% vs. 21.8%). Additionally, MRD negativity at any point during the study was observed in 60.6% of cilta-cel patients versus 15.6% of standard-care patients. Cytokine release syndrome (CRS) occurred in 76.1% of cilta-cel recipients, mostly in grades 1 or 2. CAR-T cell-related neurotoxic events were reported in 20.5% of patients, with one instance of grade 1 movement and neurocognitive adverse events. Cranial nerve palsies predominantly affecting cranial nerve VII occurred in 16 patients (9.1%), and CAR-T-related peripheral neuropathies were noted in 5 patients (2.8%). Recently, other anti-BCMA CAR-T cell products have been studied [59,60]. Orvacabtagene autoleucel (orva-cel) has been investigated in the phase 1/2 EVOLVE study (NCT03430011) for patients with RRMM: the most common adverse events were hematologic toxicities and CRS (any grade), with an ORR of 92% for all dose groups (≥VGPR: 68%) [61]. Bb21217, like ide-cel cultured with a PI3K inhibitor to enrich for T cells with memory-like phenotype, is under evaluation for heavily pretreated patients with RRMM in the ongoing first-in-human phase 1 CRB-402 trial (NCT03274219) with promising and updated safety and efficacy data for durable response [62]. FHVH33-CD8BBZ, Zevorcabtagene autoleucel (Zevor-cel, CT053), and P-BCMA-101 are novel anti-BCMA CAR-T therapies studied in phase 1/2 clinical trials [63,64,65]. Additional strategies include dual CAR-T cells targeting both BCMA and CD19 (NCT04236011; NCT04182581) [66].

Bispecific antibodies are another strategy of active immunotherapy thanks to their double antigenic specificity, which promotes cell-to-cell interaction. In fact, their mechanism of action is based on the binding between the CD3 antigen, mostly expressed by T lymphocytes, and a surface antigen expressed by MM neoplastic cells, creating an immunological synapse. This cellular interaction leads to the activation of T cells in the immune system independently of antigen presentation on the major histocompatibility complex (MHC) class 1, which releases substances, including granzyme B and perforins, subsequently initiating the apoptotic cascade within the neoplastic PC and resulting in cell death. This class of drugs has demonstrated high efficacy in terms of ORR, ranging from 65% to 79%, with a manageable safety profile [10,67,68,69,70].

The first anti-BCMA BsAb entering clinical practice is teclistamab, due to the promising results of the MajesTEC-1 phase 1/2 trial (NCT03145181, NCT04557098): at the dose of 1.5 mg/kg once a week, the ORR was 63% (≥VGPR: 59%) and the median PFS was 11 months for heavily pretreated MM patients (triple-refractory: 78%, penta-refractory: 30%), without unexpected adverse events (CRS, mostly of grade 1–2: 72%; hematological toxicity: 71%, grade ≥ 3: 65%; infections: 76%, grade ≥ 3: 45%; ICANS: 5%, grade ≥ 3: <1%) [69,70,71,72]. Recently, results from the MajesTEC-1 study’s cohort C were also reported, evaluating teclistamab in patients previously exposed to anti-BCMA therapy: once again, the responses were profound (ORR 53%, ≥VGPR: 48%), with a toxicity profile similar to that of the general study population, supporting the use of teclistamab even for patients who had relapsed after anti-BCMA targeted treatments [73]. Considering these results from the MajesTEC-1 trial, teclistamab as a single agent has been approved by the FDA and EMA for RRMM. Ongoing studies are evaluating teclistamab, both in earlier stages of the disease (NCT04722146 or MajesTEC-2, NCT05083169 or MajesTEC-3, NCT05243797 or MajesTEC-4 or EMN30, NCT05695508 or MajesTEC-5, NCT05552222 or MajesTEC-7, NCT05572515 or MajesTEC-9, NCT05849610 or GEM-TECTAL, NCT05231629 or MASTER-2, NCT05572229 or IFM2021-01, NCT05469893 or immune-PRISM) and in combination with other MoAbs, including daratumumab (NCT04108195 or TRIMM-2) and talquetamab (NCT04586426 or RedirecTT-1, TRIMM-2, NCT05338775 or TRIMM-3).

Elranatamab is another humanized IgG-like bispecific antibody targeting BCMA × CD3, evaluated in the phase 1 dose-finding and expansion MagnetisMM-1 trial (NCT03269136) in a setting of heavily pretreated patients with RRMM, showing as a single agent or in combination with lenalidomide or pomalidomide a high ORR (70%, ≥CR 30%), with a CRS rate of 83% (all grade 1–2) [74]. The phase 2 MagnetisMM-3 trial (NCT04649359) further supported the preliminary safety and efficacy data of elranatamab. This study reported results from cohort A, which included patients who had not previously received BCMA-directed therapy, totaling 123 participants [75]. The primary endpoint of a confirmed ORR was successfully achieved with an ORR of 61.0% and 35.0% of patients reaching CR or better, as assessed by a blinded independent central review. Among the responders, 50 transitioned from weekly to biweekly dosing, with 40 (80.0%) maintaining or improving their response for six months or longer. At a median follow-up of 14.7 months, the median durations of response, PFS, and OS had not yet been reached. The fifteen-month rates for the duration of response, PFS, and OS were 71.5%, 50.9%, and 56.7%, respectively. The most common adverse events included infections (69.9% any grade, 39.8% grade 3–4), CRS (57.7%, all grades 1–2), and ICANS (3.4%, all grades 1–2), highlighting a manageable safety profile. Based on these positive results, elranatamab as a single agent has been approved by the FDA and EMA for RRMM. Elranatamab is currently evaluated in several clinical studies, both in combination with other therapeutic agents and in earlier stages of the disease (NCT05090566 or MagnetisMM-4, NCT03269136, NCT05020236 or MagnetisMM-5, NCT05623020 or MagnetisMM-6, NCT05317416 or MagnetisMM-7, NCT05014412 or MagnetisMM-9, NCT05675449 or MagnetisMM-20, NCT05927571) [76].

Other anti-BCMA BsAbs under investigation are linvoseltamab (NCT03761108, NCT05730036, NCT05137054, NCT05955508), ABBV383 (NCT03933735), alnuctamab (NCT03486067) and HPN217 (NCT04184050) [77,78,79,80,81].

### 3.2. T Cell Redirecting Therapy Targeting Non-BCMA Antigens

The increasingly extensive use of anti-BCMA therapies, even in the early stages of the disease, has opened the debate on what the optimal choice should be at the time of relapse. Therefore, new alternative targets to BCMA, such as G protein-coupled receptor, family C, group 5, member D (GPRC5D), and Fc receptor homolog 5 (FcRH5), have been studied, with promising data even in the setting of patients who have already received anti-BCMA therapy (Table 2).

Novel therapeutic anti-GPRC5D strategies, such as CAR-T cells and the first-in-class BsAb talquetamab, showed great efficacy with acceptable toxicity [82,83,84,85]. Regarding talquetamab, the updated results from the phase 1/2 MonumenTAL-1 trial (NCT03399799/NCT04634552) showed a high and rapid ORR in the pivotal cohorts (74%, ≥VGPR: 59% at the dose of 405 μg/kg; 73%, ≥VGPR: 57% at the dose of 800 μg/kg), as well as in patients who had received prior T cell redirecting therapy (63%, ≥VGPR: 53%), without new safety concerns [83,86]. Based on these positive results, talquetamab as a single agent has been approved by the FDA and EMA for RRMM. Talquetamab is currently explored in further clinical studies (NCT04586426 or RedirecTT-1, TRIMM-2, NCT05050097 or MonumenTAL-2, NCT05455320 or MonumenTAL-3, NCT05461209 or MonumenTAL-5, NCT05338775 or TRIMM-3, the previously mentioned MajesTEC-7 and GEM-TECTAL). Forimtamig (RG6234) is a novel T cell-redirecting bispecific antibody designed to bind CD3 on T cells and GPRC5D on plasma cells [87,88,89]. It features a distinctive 2:1 (GPRC5D:CD3) configuration, which is theorized to enhance its potency compared to a conventional 1:1 configuration. This dual-binding mechanism facilitates the targeted destruction of plasma cells by T cells. Preclinical studies have shown that forimtamig effectively kills all tested GPRC5D+ MM cell lines. Additionally, in an autologous ex vivo model using total bone marrow aspirates from newly diagnosed MM patients, forimtamig demonstrated increased cytotoxic effectiveness. A phase 1 clinical trial (NCT04557150) is currently in progress to further investigate its efficacy and safety. The humanized bispecific antibody FcRH5 × CD3 cevostamab has been developed and studied in vivo in an ongoing phase 1 dose-finding trial (NCT03275103). Preliminary efficacy data showed promising results in heavily pre-treated patients (ORR: 57%, ≥VGPR: 33% at the target dose of 132–198 mg) with a good safety profile (CRS: 81%, mostly of grade 1–2 with early resolution within 48 h in 85% of patients) [90]. Additional studies with cevostamab, both as monotherapy and in combination with other anti-MM agents, are currently ongoing (NCT04910568 or CAMMA 1, NCT05801939, NCT05535244 or CAMMA 2, NCT05646836, NCT05583617 or PLYCOM, NCT05927571).

**Table 2 cells-13-00853-t002:** Results of the main clinical trials of T cell-redirecting immunotherapy.

	Ide-Cel [52,57]	Cilta-Cel [53,54,58,59,60]	Teclistamab [69,70,71,72,73]	Elranatanab [75]	Talquetamab [83,86]	Cevostamab [90]
**Trial (phase)**	KarMMa (2) KarMMa-3 (3)	CARTITUDE-1 (1b/2) CARTITUDE-4 (3)	MajesTEC-1 (1/2)	MagnetisMM-3 (2)	MonumenTAL-1 (1/2)	NCT03275103 (1)
**Target**	BCMA	BCMA	BCMA	BCMA	GPRC5D	FcRH5
**No. of pts**	128 386	97 419	165	123	232	160
**No. of prior lines**	≥3 2–4	≥3 1–3	≥3	5 (median)	6 (median)	6 (median)
**Median f. up (mo)**	13 19	28 16	14	15	12 (405-μg) * 4 (800-μg)	6
**ORR (%)**	73 71	98 85	63	61	70 (405-μg) 64 (800-μg)	57
**CR or better (%)**	33 39	83 73	39	35	23 22	33 §
**MRD-negativity (%)**	26 20	92 61	27	90	/	/
**PFS (mo)**	9 13	NR NR	11	NR	48% (405-μg) ^ 68% (800-μg)	/
**Hematological toxicity (%)** ** *grade ≥ 3* **	91 90 *89* *87*	96 95 *95* *94*	71 *64*	49 *49*	67 *60*	18 ^X^ - *16*
**Infections (%)** ** *grade ≥ 3* **	25 58 *2* *24*	21 62 *none* *27*	76 *45*	70 *40*	47 (405-μg) ^ 34 (800-μg) *7 in both cohorts*	43 - *19*
**CRS (%)** ** *grade ≥ 3* **	84 88 *5* *4*	95 76 *4* *1*	72 - *1*	58 *none*	77 (405-μg) ^ 80 (800-μg) *3* (405-μg) *none* (800-μg)	80 - *1*
**ICANS (%)** ** *grade ≥ 3* **	18 15 *<3* - *3*	17 5 ^W^ *<3* - *none*	3 *none*	0 *none*	10 (405-μg) ^ 5 (800-μg) *none in both cohorts*	16 ^Y^ - *1*
**Skin-related (%)** ** *grade ≥ 3* **	NA	NA	NA	NA	67 (405-μg) ^ 70 (800-μg) *none* (405-μg) *2* (800-μg)	NA
**Dysgeusia (%)** ** *grade ≥ 3* **	NA	NA	NA	NA	63 (405-μg) ^ 57 (800-μg) *none in both cohorts*	NA

No.: number; pts: patients; f. up: follow up; mo: months; ORR: overall response rate; CR: complete remission; MRD: minimal residual disease; PFS: progression-free survival; NR: not reached; CRS: cytokine release syndrome; ICANS: neurological toxicity associated with effector cells of the immune system; NA: not applicable. * two cohorts with different dose levels: 405-μg dose level and 800-μg dose level; § rate of very good partial response or better; ^ percentage of patients who were alive and progression-free at the time of data cutoff; ^X^: rate of neutropenia; ^W^: 20.5% of pts infused with cilta-cel experienced neurotoxicity different from ICANS, mostly of grade 1–2; ^Y^: ICANS was observed in CRS events.Finally, other novel cellular therapies targeting molecules different from BCMA, including CD38 (NCT03464916), CD44v6 (NCT04097301), CD138 (NCT03672318), and SLAMF7 (NCT03958656), are under investigation.

## 4. Discussion

Currently, ASCT remains the standard of care for NDMM patients eligible for high-dose chemotherapy. Improved response rates and PFS were obtained thanks to the incorporation of anti-CD38 monoclonal antibodies in the frontline setting. The striking efficacy of the new T cell redirecting therapies has raised the question of where to place them in the treatment algorithm of NDMM. About this debate, there are several ongoing studies evaluating the role of ASCT in the era of immunotherapy (Table 3). Particularly, these clinical trials should answer some open issues: will ASCT remain the first-line choice for young and fit patients compared to CAR-T cell therapy? Will integrating immunotherapy Yes I agree into transplantation, such as consolidation with CAR-T cells or maintenance with BsAbs, increase the rate of MRD-negativity and, consequently, improve survival?

The phase 1 KarMMa-4 trial (NCT04196491) explores the optimal target dose and the safety profile of ide-cel in high-risk NDMM patients. The two phase 3 studies, CARTITUDE-5 (NCT04923893) and CARTITUDE-6 (NCT05257083) are planned to evaluate the efficacy of cilta-cel in patients with NDMM. Particularly, the CARTITUDE-6 trial compares CAR-T cell therapy versus ASCT after induction with D-VRd in transplant-eligible NDMM patients. Another phase ½ study (NCT01352286) is evaluating the role of engineered T cell immunotherapy as a consolidation strategy after ASCT for high-risk MM patients [91].

The phase 2 MajesTEC-5 trial (NCT05695508) is recruiting NDMM patients to evaluate the safety and tolerability of teclistamab both as induction therapy in combination with VRd and D-VRd and as maintenance therapy post-ASCT. The phase 3 MajesTEC-4 study (NCT05243797 or EMN30) is exploring the use of teclistamab as maintenance after ASCT. Among the MRD-driven multi-cohort MASTER-2 study (NCT05231629) MRD-positive NDMM patients will receive teclistamab plus daratumumab as post-ASCT consolidation and maintenance. Again, the ongoing phase 3 MagnetisMM-7 study (NCT05317416) investigates elranatamab versus lenalidomide as maintenance after ASCT [92]. The phase 2 GEM-TECTAL (NCT05849610) study will assess the efficacy and safety of teclistamab and talquetamab combined with daratumumab in de novo high-risk MM in both transplant-eligible and -ineligible patients.

## 5. Conclusions

The field of immunotherapy in MM is rapidly evolving, with an impressive number of ongoing clinical trials. Considering the latest data, active immunotherapy with CAR-T cells and BsAbs will soon revolutionize the therapeutic landscape of MM; thus, the state-of-the-art frontline treatment, including the transplant program, is expected to change in the next few years. Indeed, considering on one hand both the acute toxicities of HDM, such as hematologic and gastrointestinal toxicity, infections, and the long-term toxicities, such as secondary neoplasms [93], and considering also the significant increase in deep responses with quadruplet induction therapy and the great results of CAR-T cells and BsAbs treatment (see Table 2), it is likely that in the near future HDM-ASCT will be deferred or reserved for selected cases, leading to a risk- and a response-adapted therapy, improving the patient’s quality of life and survival.

## Figures and Tables

**Table 3 cells-13-00853-t003:** T cell-redirecting therapy in the front-line setting in transplant-eligible MM patients.

Trial	NCT Number	Phase	Patients	Design
**KarMMa-4**	NCT04196491	1	HR NDMM, TE	bb2121 autologous CAR T cells + R as maintenance
**CARTITUDE-5**	NCT04923893	3	NDMM, ASCT is not planned ^a^	VRd + cilta-cel vs VRd + Rd
**CARTITUDE-6**	NCT05257083	3	NDMM, TE	DVRd + cilta-cel vs DVRd + ASCT
**NCT01352286**	NCT01352286	1/2	HR NDMM, TE ^b^	anti-CD3/anti-CD28-costimulated autologous T cells after ASCT ^c^
**MajesTEC-5**	NCT05695508	2	NDMM, TE	Teclistamab + DRd ± V as induction and Teclistamab + DR as maintenance
**MajesTEC-4 (EMN30)**	NCT05243797	3	NDMM, TE	Teclistamab + R vs Teclistamab vs R as maintenance after ASCT
**MASTER-2**	NCT05231629	2	NDMM, TE, MRD positive	Teclistamab + D as consolidation and maintenance after ASCT
**MagnetisMM-7**	NCT05317416	3	NDMM, TE	Elranatamab vs R as maintenance after ASCT
**GEM-TECTAL**	NCT05849610	2	HR NDMM, TE, or NTE	Teclistamab + Talquetamab + D

HR: high-risk; NDMM: newly diagnosed multiple myeloma; TE: transplant eligible; R: lenalidomide; ASCT: Autologous Stem Cell Transplant; VRd: Bortezomib, Lenalidomide, and Dexamethasone; cilta-cel: Ciltacabtagene Autoleucel; Rd: Lenalidomide and Dexamethasone; DVRd: Daratumumab, Bortezomib, Lenalidomide, and Dexamethasone; DRd: Daratumumab, Lenalidomide, and Dexamethasone; V: Bortezomib; DR: Daratumumab and Lenalidomide; MRD: minimal residual disease; D: Daratumumab; NTE: non-transplant eligible. ^a^: Not considered for high-dose chemotherapy with ASCT due to: ineligible due to advanced age or Ineligible due to the presence of comorbid condition(s) likely to have a negative impact on the tolerability of high-dose chemotherapy with ASCT or Deferral of high-dose chemotherapy with ASCT as initial treatment. ^b^: HLA-A201 patients must have confirmed expression of NY-ESO-1 and/or LAGE. HLA-A2 patients must have the A-201 allele. ^c^: T cells that have been genetically modified to express high affinity NY-ESO-1c259 TCRs.

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
