# Peer review of "Multiple Myeloma: The Role of Autologous Stem Cell Transplantation in the Era of Immunotherapy"

_cells, 2024, doi:10.3390/cells13100853_

Round 1

Reviewer 1 Report

Comments and Suggestions for Authors

This is a well-written review whose aim is to explore the role of autologous HCT for MM in the era of immunotherapy. The references are clear, the data are accurate, and the flow of the manuscript is logical.

My only major criticism is that, there is no major conclusion to be drawn from the manuscript. The authors diligently repeat data from the major trials, yet do not offer any suggestions about how immunotherapy might change the treatment landscape for MM, besides discussing briefly some trials that will be conducted.

This manuscript would be strengthened by making clearer statements about what the authors think will happen with AHCT in MM. Many in the US are of the opinion that eventually BCMA CAR T will replace AHCT in the frontline setting for MM due to better patient tolerability and interest levels. There are already some preliminary data suggesting that QOL metrics are superior with CAR T vs AHCT. The manuscript should ideally have a section devoted to discussion of this and state the authors opinion - not buried in the conclusion. As currently written, it is just a review of the literature – please state your opinions and cite some data to support.

If word count is too high, consider removing the section on stem cell mobilization – why is this needed? It really is not relevant to the main topic of the review.

Minor points:

1.page 1 line 24 – should be “plasma cell” not “plasmacell” – this error is present multiple places in the manuscript, please change.

2. Page 1 line 30, consider saying “… continues to be the standard of care for eligible patients with newly diagnosed MM>..” – age is not an absolute contraindication for AHCT globally, esp in the US

3. Page 1, lines 42-45 – not sure why the authors inserted a discussion on carcinogenesis, it really doesn’t fit, please either clarify or remove.

Author Response

This is a well-written review whose aim is to explore the role of autologous HCT for MM in the era of immunotherapy. The references are clear, the data are accurate, and the flow of the manuscript is logical.

  • Thank you very much for your appreciation. We followed your suggestions to improve the manuscript.

My only major criticism is that, there is no major conclusion to be drawn from the manuscript. The authors diligently repeat data from the major trials, yet do not offer any suggestions about how immunotherapy might change the treatment landscape for MM, besides discussing briefly some trials that will be conducted. This manuscript would be strengthened by making clearer statements about what the authors think will happen with AHCT in MM. Many in the US are of the opinion that eventually BCMA CAR T will replace AHCT in the frontline setting for MM due to better patient tolerability and interest levels. There are already some preliminary data suggesting that QOL metrics are superior with CAR T vs AHCT. The manuscript should ideally have a section devoted to discussion of this and state the authors opinion - not buried in the conclusion. As currently written, it is just a review of the literature – please state your opinions and cite some data to support.

  • It is a constructive criticism. We tried to reformulate the conclusion, emphasizing our opinion, as follow:

“The field of immunotherapy in MM is rapidly evolving with an impressive number of ongoing clinical trials. Considering the latest data, active immunotherapy with CAR-T cells and BsAbs will soon revolutionize the therapeutic landscape of MM, thus the state of the art of frontline treatment, including the transplant program, is expected to change in the next few years. Indeed, considering on one hand both the acute toxicities of HDM, such as hematologic toxicity, infections, gastrointestinal toxicity, and the long-term toxicities, such as secondary neoplasms [93], and considering also the significant increase in deep responses with quadruplet induction therapy and the great results of CAR-T cells and BsAbs treatment (see Table 2 and Table 3), it is likely that in the near future HDM-ASCT will be deferred or reserved for selected cases, leading to a risk- and a response-adapted therapy, improving the patients’ quality of life and survival.” 

If word count is too high, consider removing the section on stem cell mobilization – why is this needed? It really is not relevant to the main topic of the review.

  • Thank for the suggestion. We would like to provide a brief summary of recent data on stem cell collection as well, given the discussion on ASCT, but following your advice, we have reduced the length pf this chapter.

Minor points:

1.page 1 line 24 – should be “plasma cell” not “plasmacell” – this error is present multiple places in the manuscript, please change.

  • Sorry for this typo. We corrected it in all the manuscript.

  1. Page 1 line 30, consider saying “… continues to be the standard of care for eligible patients with newly diagnosed MM>..” – age is not an absolute contraindication for AHCT globally, especially in the US

- It is a good consideration. Therefore, I change the phrase like your suggestion:

“…continues to be the standard of care for eligible patients with newly diagnosed MM..”

  1. Page 1, lines 42-45 – not sure why the authors inserted a discussion on carcinogenesis, it really doesn’t fit, please either clarify or remove.

  • We included a chapter on carcinogenesis to emphasize the importance of the patient's immune system not only in solid tumors but also in hematologic neoplasms such as MM (a concept of recent interest). However, upon reading the text, I agree that it doesn’t fit, so I removed it.

Reviewer 2 Report

Comments and Suggestions for Authors

In this manuscript, the authors reviewed the incorporation of immunotherapy in the treatment scenario of patients with newly diagnosed multiple myeloma (NDMM) that are eligible to autologous stem cell transplantation (ASCT). This review is very interesting and well written but still needs improvement. 

1. In terms of T-cell redirecting therapy in MM targeting BCMA including CAR-T and BsAbs (Part 3.1), it would be better to make a table to summarize the results (ORR and/or PFS) and toxicities (adverse events) from different clinical trials for the convenience of readers. The same is true for the Part 3.2.

2.The references in all tables should be added, if applicable.

3. There are some typos, for example, Line 15, there should be a space between “MM” and “is”; Line 70, there should be a dash between “anti” and “CD38”, etc. Please check the whole manuscript carefully.

4. The abbreviation of proteasome inhibitor(s) is provided twice in the manuscript. 

Author Response

In this manuscript, the authors reviewed the incorporation of immunotherapy in the treatment scenario of patients with newly diagnosed multiple myeloma (NDMM) that are eligible to autologous stem cell transplantation (ASCT). This review is very interesting and well written but still needs improvement.

  • Thank you for your kind comment. The paper has been improved with your suggestions.

  1. In terms of T-cell redirecting therapy in MM targeting BCMA including CAR-T and BsAbs (Part 3.1), it would be better to make a table to summarize the results (ORR and/or PFS) and toxicities (adverse events) from different clinical trials for the convenience of readers. The same is true for the Part 3.2.

  • Thank you for this suggestion. We added a table (Table 2) in which we summarized the results of efficacy and safety of the main clinical trials of T-cell redirecting therapy (part 3.1 and part 3.2)

2.The references in all tables should be added, if applicable.

  • As you suggested, we added references in table 1 and table 2; for table 3 this is not applicable because results of trials are ongoing

  1. There are some typos, for example, Line 15, there should be a space between “MM” and “is”; Line 70, there should be a dash between “anti” and “CD38”, etc. Please check the whole manuscript carefully.

  • Sorry for these typos. We corrected them and we check carefully all the manuscript.

  1. The abbreviation of proteasome inhibitor(s) is provided twice in the manuscript.

  • Thank you, we remove the second one and other repetitions.

Round 2

Reviewer 1 Report

Comments and Suggestions for Authors

The authors have satisfied all my questions/concerns.